# Mechanisms of Deamidation of Asparagine Residues and Effects of Main-Chain Conformation on Activation Energy

**DOI:** 10.3390/ijms21197035

**Published:** 2020-09-24

**Authors:** Koichi Kato, Tomoki Nakayoshi, Eiji Kurimoto, Akifumi Oda

**Affiliations:** 1College of Pharmacy, Kinjo Gakuin University, 2-1723 Omori, Moriyama-ku, Nagoya, Aichi 463-8521, Japan; 2Faculty of Pharmacy, Meijo University, 150 Yagotoyama, Tempaku-ku, Nagoya, Aichi 468-8503, Japan; 184331503@ccmailg.meijo-u.ac.jp (T.N.); kurimoto@meijo-u.ac.jp (E.K.); oda@meijo-u.ac.jp (A.O.); 3Institute for Protein Research, Osaka University, 3-2 Yamadaoka, Suita, Osaka 565-0871, Japan

**Keywords:** deamidation, molecular dynamics simulation, quantum chemical calculation, post-translational modification, age-related diseases

## Abstract

Deamidation of asparagine (Asn) residues is a nonenzymatic post-translational modification of proteins. Asn deamidation is associated with pathogenesis of age-related diseases and hypofunction of monoclonal antibodies. Deamidation rate is known to be affected by the residue following Asn on the carboxyl side and by secondary structure. Information about main-chain conformation of Asn residues is necessary to accurately predict deamidation rate. In this study, the effect of main-chain conformation of Asn residues on deamidation rate was computationally investigated using molecular dynamics (MD) simulations and quantum chemical calculations. The results of MD simulations for γS-crystallin suggested that frequently deamidated Asn residues have common main-chain conformations on the N-terminal side. Based on the simulated structure, initial structures for the quantum chemical calculations were constructed and optimized geometries were obtained using the B3LYP density functional method. Structures that were frequently deamidated had a lower activation energy barrier than that of the little deamidated structure. We also showed that dihydrogen phosphate and bicarbonate ions are important catalysts for deamidation of Asn residues.

## 1. Introduction

Deamidation of asparagine (Asn) residue is a post-translational modification observed in various proteins and has been reported to spontaneously and nonenzymatically occur in vivo and in vitro under physiological conditions [1,2,3,4,5,6]. This reaction typically proceeds through the cyclic succinimide intermediate, resulting in the production of Asp and biologically uncommon isoAsp residues in an approximately 1:3 molar ratio (Figure 1) [7]. These conversions from Asn residues to acidic residues by deamidation affect the structural stability of proteins and are seriously disruptive to biologically important functions [5,8,9,10,11,12]. Some of the deamidated proteins are targeted for degradation by the ubiquitin–proteasome system [13]. Therefore, deamidation is hypothesized to serve as a molecular clock for biological processes such as protein turnover. In addition, because deamidation can result in protein denaturation and aggregation, it is suggested to trigger several age-related diseases [14,15,16,17,18]. In fact, deamidated residues are frequently detected in crystallins isolated from the eye lens of patients suffering from age-related nuclear cataracts [16,17,18]. Furthermore, isoAsp formation in vivo is involved in autoimmune diseases [19,20,21,22]. An isoAsp-containing peptide triggers an autoimmune response to the native peptide containing the original Asn residue. For instance, isoAsp formation in histone H2B has been suggested to be involved in systemic lupus erythematosus [21,22]. Furthermore, Asn deamidation plays a key role in quality control of antibody drugs. Deamidations have been reported to occur in complementarity-determining regions (CDRs) and Fc regions of recombinant monoclonal antibodies and human endogenous IgG in vivo [23,24,25,26,27]. In particular, antigen-binding affinities are reduced by Asn deamidation in CDRs. Therefore, Asn deamidation is an important reaction for not only pathological analysis but also development of peptide/protein-based drugs.

Asn deamidation rate is affected by various factors, especially adjacent residues on the C-terminal side ((*N* + 1) residue) [28,29,30]. For instance, the deamidation half-time for the Asn–Gly sequence is approximately 300-fold smaller than that for the Asn–Ile sequence in a pentapeptide [29]. In contrast, the half-time for the Asn–His sequence is approximately 9-fold longer than that for the Asn–Gly sequence. In the Asn–Pro sequence, for which succinimide cannot be formed, the half-time is 7000-fold longer than that for the Asn–Gly sequence in a pentapeptide. Other important factors reported to affect Asn-deamidation rates are secondary structure, tertiary structure, quaternary structure, hydrogen-bond formation, surface exposure, and solvent molecules [5,7,10,31,32]. Based on this information, some predictive methods for Asn-deamidation rates in proteins have been proposed [33,34,35,36,37,38]. A predictive method produced by Robinson and Robinson is currently considered to be the most reliable method [33,34]. In this method, amino acid sequence, secondary structures, and hydrogen bonds with other residues are used for the prediction. These factors are involved in succinimide formation from Asn residues. Succinimide formation is considered to be caused by nucleophilic attack of the amide nitrogen of the (*N* + 1) residue to the amide carbon of the Asn side chain, and proceeds via the tetrahedral intermediate that includes *gem*-hydroxylamine (Figure 2). Structural parameters involved in the above nucleophilic attack are expected to be important factors for deamidation rates [35,36,37,38]. The distance between the main-chain nitrogen of the (*N* + 1) residue and amide carbon of the Asn side chain (N*_N_*_+1_–C_γ_ distance) in crystal structures is used to predict deamidation rate in monoclonal antibodies [35,37,38]. Furthermore, these methods use dihedral angles of Asn residues in crystal structures as a parameter for predicting deamidation rates. In cyclic peptides containing the Asn–Gly sequence, the deamidation rate is higher in the peptide with the shorter N*_N_*_+1_–C*_γ_* distance [39,40]. In addition, deamidation is facilitated when the N*_N_*_+1_–C*_γ_*–O*_γ_* angle is <128°. However, deamidation rate is low in a cyclic peptide satisfying these parameters (Lys–Asn–Gly–Arg–Glu–NH_2_). Therefore, although these parameters are key in determining deamidation rate, other factors are also important. Those parameters might be difficult to apply to the flexible side chain of Asn residues in protein structures. Therefore, the relevance for deamidation rate of the N*_N_*_+1_–C*_γ_* distance and dihedral angles in crystal structures is unclear for calculations of protein dynamics.

Some computational studies have reported the succinimide formation mechanism of Asn and the changes of dihedral angles with the deamidation reaction using quantum chemical calculations [41,42,43,44]. The structural changes with succinimide formation from Asn are relatively small (change of N‒C_α_‒C‒N and N‒C_α_‒C_β_‒C_γ_: 35–50°, C‒N‒C_α_‒C: nearly constant) in comparison with other nonenzymatic modifications in some previous studies (the maximum change of dihedral angles: 214°) [45,46,47,48,49]. Therefore, the flexibilities of the main chain are assumed not to be a primary factor in determining deamidation rate. However, previous computational studies of nonenzymatic modification of amino acids have not considered protein structure. In this study, conformations of Asn residues in a protein were first investigated to analyze the influence of three-dimensional (3D) structure on the Asn deamidation rate. We focused on the 3D structure of γS-crystallin, which is targeted for deamidation analysis in some previous studies [32,50,51,52]. The experimental structure registered in the protein data bank (PDB) was obtained, and Asn conformations in γS-crystallin were investigated. Furthermore, molecular dynamics (MD) simulations were performed to evaluate conformational flexibilities of Asn residues. Second, the reaction mechanisms and activation energy barriers in succinimide formation from Asn residues were investigated using quantum chemical calculations. Two different initial structures were constructed based on the Asn conformations in the experimental and calculated structures. As the deamidation rate in the phosphate buffer is higher than that in Tris–HCl buffer [34], a dihydrogen phosphate ion was employed as the catalyst. To compare mechanisms and activation energy barriers, water-catalyzed and carbonate-catalyzed reactions were also analyzed.

## 2. Results and Discussion

### 2.1. Structural Features of Frequently Deamidated Asn Residues

To investigate the effects of Asn conformation on deamidation rate, all Asn residues in the γS-crystallin structure were examined (Figure 3). The level of deamidation at Asn14, Asn53 and Asn143 is high [32]. Notably, approximately 60% of Asn14 is reported to be deamidated in water-insoluble γS-crystallin extracted from eye lenses of patients with age-related nuclear cataracts. Namely, these Asn residues are frequently deamidated residues. In contrast, Asn37 and Asn76 are infrequently deamidated: their deamidation ratios are 15% and 8%, respectively. Dihedral angles of all Asn residues (Figure 4) in γS-crystallin are shown in Table 1. The dihedral angles for H–N–C_α_–H, defined as *φ*_H_, of frequently deamidated residues were −13.2°, 19.4° and 10.4°, respectively. They are therefore syn periplanar. On the other hand, Asn37 was anti periplanar, and Asn76 was neither. Here, the conformations of Asn14, Asn53 and Asn143 were defined as syn conformations, and that of Asn37 was defined as anti conformation. Asn residues with syn conformations are considered to have extensive deamidation. The main-chain conformations of Asn residues may be important for the deamidation rate. To analyze protein dynamics, a 2000-ns MD simulation was performed for the γS-crystallin structure. The root mean square deviation (RMSD) plot is shown in Figure 5. The RMSD value at the end of the simulation was approximately 3.0 Å, and a large structural change was not observed. The dihedral angles of all Asn residues in the final structure of the simulation are shown in Table 2. Asn14 and Asn143 formed syn conformations, and Asn37 and Asn76 formed anti conformations in the final structure. Although Asn53 was not a syn conformation in the final structure, all frequently deamidated Asn residues were syn conformations for most of the final 10 ns of simulation (Table 3). Asn37 and Asn76 formed anti conformations throughout the simulation. Dihedral angles *φ* in the frequently deamidated residues were about 55° throughout the simulation (Appendix A). The values of *χ* were not constant through the last 10 ns of the trajectories for all Asn residues. The dihedral angles *ψ* of frequently deamidated residues were about 40°. Those of the infrequently deamidated residues Asn37 and Asn76 were about 20° and 150°, respectively. Therefore, dihedral angles *φ*, *ψ*, and *φ*_H_ were common in frequently deamidated residues; however, only *φ*_H_ was common in infrequently deamidated residues.

To evaluate the influence of backbone flexibility on the deamidation rate, the root mean square fluctuation (RMSF) values of the final 10 ns trajectories were calculated (Figure 6A). As the highest RMSF value was 1.60 Å at Ile160, the main chain of γS-crystallin was not highly flexible. Relatively flexible regions (RMSF > 1.00 Å) are indicated in the whole structure after MD simulations (Figure 6B). No Asn residues were located on the flexible regions. Although Asn76 was the most flexible Asn residue, it is infrequently deamidated. Therefore, the flexibility of the main chain is considered to have little effect on Asn deamidation.

### 2.2. Deamidation Mechanism in the Syn Conformation

To investigate the reaction mechanism and activation energy barrier, quantum chemical calculations were performed. The model compound used in this study was Asn capped with acetyl (Ac) and methylamino (NMe) groups on the N- and C-termini, respectively (Figure 7). Two H_2_O molecules and one dihydrogen phosphate (H_2_PO_4_^−^) ion or one bicarbonate (HCO_3_^−^) ion was included in the calculations as catalysts. These molecules and ions abundantly exist in vivo and have been suggested to catalyze post-translational modification of proteins [41,42,43,44,45,46,47,48,49]. The optimized geometries of the cyclization step, which occur by nucleophilic attack by the (*N* + 1) amide nitrogen to carbonyl carbon of the side chain, for the syn conformation in the phosphate-catalyzed reaction are shown in Figure 8. Those of water-catalyzed and carbonate-catalyzed reactions are shown in Appendix A. In the reactant complex (RC), three hydrogen bonds were formed between the Asn residue and the H_2_PO_4_^−^ ion (Figure 8). A hydrogen bond between the amide oxygen of the (*N* − 1) residue and the H_2_PO_4_^−^ ion was not involved in the cyclization reaction, while it might stabilize the conformation of RC. In addition to this hydrogen bond, the amide NH hydrogen of (*N* + 1) residue, and amide oxygen of the side chain formed hydrogen bonds with the H_2_PO_4_^−^ ion. The cyclization step was started by proton transfer between these hydrogen-bonding atoms. Proton transfer between the (*N* + 1) amide nitrogen and the H_2_PO_4_^−^ ion was almost completed in the transition state 1 (TS1). In contrast, the proton migrating from the H_2_PO_4_^−^ ion to amide oxygen of side chain was located between those atoms. Therefore, the (*N* + 1) amide hydrogen is considered to be abstracted by the H_2_PO_4_^−^ ion early in the cyclization step, and this proton abstraction is expected to enhance the nucleophilicity of amide nitrogen. Along with those proton transfers, the distance between the (*N* + 1) amide nitrogen and the amide carbon of side chain was shortened in TS1 (2.20 Å). The completion of those proton transfers and the five-membered ring formation resulted in a tetrahedral intermediate 1 (INT1) formation. The nitrogen of the (*N* + 1) amide formed no hydrogen bond with the H_2_PO_4_^−^ ion. In contrast, the H_2_PO_4_^−^ ion formed two hydrogen bonds with the (*N* − 1) amide oxygen and OH hydrogen of the *gem*-hydroxylamine moiety. A hydrogen bond between the (*N* − 1) amide oxygen and the H_2_PO_4_^−^ ion was maintained throughout the reaction. Proton transfer for the cyclization step in the water- and carbonate-catalytic reactions was similar to that in the phosphate-catalytic reaction (Appendix A). Although the hydrogen bond between the HCO_3_^−^ ion and (*N* − 1) amide oxygen was formed in INT1, it was not observed in RC and TS1. In the carbonate-catalyzed reaction, stabilization of the complexes by their interaction between the Asn residue and the HCO_3_^−^ ion was relatively weak.

The optimized geometries in the deammoniation step of the phosphate-catalyzed reaction are shown in Figure 9. A tetrahedral intermediate 2 (INT2) was formed by rearrangement of the H_2_PO_4_^−^ ion. The position of the H_2_PO_4_^−^ ion and the conformation of the *gem*-hydroxylamine moiety were considered to be easily changed because these ions are abundant under physiological conditions. In INT2, the H_2_PO_4_^−^ ion formed two hydrogen bonds with the nitrogen and OH hydrogen atoms of the *gem*-hydroxylamine moiety (1.77 Å and 1.63 Å, respectively). The proton transfers between these hydrogen-bonded atoms resulted in TS2 formation (Figure 9). In TS2, these proton transfers were almost completed, and the C‒N distance in *gem*-hydroxylamine moiety was stretched by 0.37 Å. In the product complex (PC), the C‒N distance was 4.26 Å. Therefore, the C‒N bond was cleaved after completion of proton transfer. Proton transfers in the water-catalyzed reaction are not completed in the TS of the deammoniation step (Appendix A). In contrast, the C‒N bond was cleaved after completion of proton transfer in the carbonate-catalyzed reaction (Appendix A). The protonation of the nitrogen atom of *gem*-hydroxylamine has been suggested to occur in the INT2 [43,44], and its proton transfer is considered to precede the C–N bond cleavage. Our results of the intrinsic reaction coordinate (IRC) calculation also suggested that the nitrogen atom of *gem*-hydroxylamine is protonated in the early stage of TS2 formation in the reaction catalyzed by water, H_2_PO_4_^−^ and HCO_3_^−^ ions.

The changes of dihedral angles through the cyclization step in the phosphate-catalyzed reaction are shown in Table 4. There were no significant changes of *φ* in all steps, and *φ* remained around 55° from TS1 to PC. The change of *ψ* in conversion from RC to TS1 was the largest change during the reaction (31.2°). These changes of dihedral angles were relatively smaller than that for succinimide formation from Asp [46,47,49]. Succinimide formation from Asn causes small conformational changes in comparison with its formation from Asp. As φ_H_ was constant, the syn conformation was maintained throughout the cyclization and deammoniation steps. The changes of dihedral angles in the water-catalyzed reaction were similar to those in the phosphate-catalyzed reaction (Appendix A). In contrast, the alteration of the dihedral angle *ψ* in the carbonate-catalyzed reaction was relatively small in TS1 formation from RC (17.1°). In the PC of all reactions, *φ* and *ψ* are approximately 55° and 140°, respectively. Dihedral angles of *φ* and *φ*_H_ in the optimized geometries for the succinimide formation were consistent with those in the Asn residues in the structure of γS-crystallin obtained by the MD simulation. Therefore, dihedral angles of *φ* and *φ*_H_ can be used to predict the Asn deamidation rate. On the other hand, *ψ* and *χ* were different from those of frequently deamidated residues in the PDB and MD simulations. These dihedral angles are assumed to not be important for deamidation rate.

### 2.3. Deamidation Mechanism in the Anti Conformation

For comparison of the Asn deamidation mechanism between syn and anti conformation, quantum chemical calculations for the anti conformation were also performed. The optimized geometries of the cyclization step in the phosphate-catalyzed reaction are shown in Figure 10. Overall, proton transfer and nucleophilic attack for cyclization were similar to those in the syn conformation. In addition, proton abstraction by the H_2_PO_4_^−^ ion from the (*N*+1) amide nitrogen occurred at an early stage of the cyclization step. However, the H_2_PO_4_^−^ ion formed a hydrogen bond with the amide NH proton of the N-terminal side. The reaction mechanism of the anti conformation in water- and carbonate-catalyzed reactions was also similar to those of the syn conformation (Appendix A). H_2_O molecules formed no hydrogen bond with the N-terminal main chain atoms in the water-catalyzed reaction. On the other hand, differences in hydrogen bonds between optimized geometries of syn and anti conformations were not observed in the carbonate-catalyzed reaction. Differences in hydrogen-bond formation might affect the activation energy barrier.

The optimized geometries for the deammoniation step in the phosphate-catalyzed reaction are shown in Figure 11. As in the cyclization step, the reaction mechanism of the deammoniation step in the anti conformation was similar to that in the syn conformation. There are no differences in hydrogen-bond formation in the deammoniation step between the syn and anti conformations. In addition, distances between the atoms related to reactions in the anti conformation were almost the same as those of the syn conformation. Therefore, the syn/anti conformations do not affect the mechanism of the deammoniation step. In the water- and carbonate-catalyzed reactions, the deammoniation mechanisms for the anti conformation are considered to be similar to those for the syn conformation (Appendix A).

The changes in dihedral angles of the main chain in the anti conformation were larger than those of the syn conformation (Table 5). The amount of alteration for *φ* and *ψ* from RC to TS1 was 28.1° and 86.8°, respectively. *χ* was nearly constant throughout the whole reaction. Therefore, large structural changes of the main chain were required for succinimide formation in the anti conformation compared with those in the syn conformation. This is considered to be one of the reasons for the low frequency of deamidation in the anti conformation. In addition, Asn37 and Asn76 of γS-crystallin rarely formed suitable conformations for succinimide formation (Appendix A). Therefore, structural constraints are considered to result in Asn37 and Asn76 rarely being deamidated.

### 2.4. Activation Energy for Succinimide Formation from the Asn Syn/Anti Conformation

To evaluate the differences in activation energy barrier between the syn and anti conformations, the deamidation pathways of Asn residues were investigated for syn and anti conformations using quantum chemical calculations. The relative energies were calculated using the MP2/6-311+G(2d,2p)//B3LYP/6-31+G(d,p) level (Figure 12 and Appendix A). The energy profiles for two-step succinimide formation from an Asn residue in the phosphate-catalyzed reaction are shown in Figure 12A, B. In both conformations, the relative energy of TS1 was higher than that of TS2. Therefore, the cyclization steps are considered to be the rate-determining steps. The overall activation energies in the syn and anti conformations were 89.3 and 111 kJ·mol^−1^, respectively. Although the activation energy barriers in both conformations were significantly low to allow the deamidation reaction to occur under physiological conditions, the barrier of the syn conformation was lower than that of the anti conformation. The activation energy barriers of the syn conformation were in agreement with those experimentally determined for deamidation of model peptides in phosphate buffer at pH 7.5 (90.7 kJ·mol^−1^) [7]. This result suggests that succinimide formation from Asn in the syn conformation occurs more easily than in the anti conformation, consistent with experimental data for the deamidation level in γS-crystallin [32]. Furthermore, the activation energies of the syn conformation in carbonate-catalyzed deamidation was 82.7 kJ·mol^−1^ (Figure 12C, D). This value was also consistent with activation barriers obtained by experimental methods in various buffers (80–100 kJ·mol^−1^) [6,7]. Therefore, H_2_PO_4_^−^ and HCO_3_^−^ ions are considered to be major catalysts for Asn deamidation.

## 3. Methods

### 3.1. Investigation of Asn Conformation in the Experimental Structure

To investigate the influence of the 3D structure on Asn deamidation rates, an experimental structure of γS-crystallin [53] was obtained from the PDB (PDB ID: 2M3T). To retrieve common structural features of frequently deamidated Asn residues, the Asn conformations were observed using the PyMOL molecular viewer (http://pymol.sourceforgen.net/).

### 3.2. Molecular Dynamics Simulations

To consider structural flexibility, the MD simulation was performed for the γS-crystallin structure. The system was solvated with the TIP3P model [54], neutralized by adding Cl^−^ ion, and calculated under the periodic condition. The cutoff distance of van der Waals interactions was set at 10 Å, and the particle mesh Ewald method [55] was used for calculating electrostatic interactions. The SHAKE algorithm [56] was applied to constrain the lengths of bonds containing hydrogen atoms. Before MD simulations, 1000-step structural minimizations of water and counter ion were performed, and 2500-step minimizations of whole systems were performed, followed by temperature-increasing MD simulation of 20 ps with the temperature raised from 0 to 300 K. After these simulations, equilibrating MD simulation was carried out under constant temperature and pressure for 2000 ns. The time steps of temperature-increasing and equilibrating MD simulations were 2 fs. These MD simulations were accomplished using AMBER16 [57]. AMBER ff14SB force field [58] was used for the parameters of amino acids. The cpptraj module of AmberTools17 was used for analysis of the results. The root mean square deviations (RMSDs) for main-chain atoms are calculated using the 3D structures after temperature-increasing MD simulations. The root mean square fluctuations (RMSFs) were calculated for all C_α_ atoms using the MD trajectories for the last 10 ns to evaluate the structural flexibilities. As a reference structure for RMSF calculations, the average structures of the last 10 ns trajectories were used. The dihedral angles *φ*, *ψ*, *χ*, and *φ*_H_ were investigated using the last 10 ns trajectories.

### 3.3. Quantum Chemical Calculations

Based on the experimental data and the calculated structures of Asn residues in γS-crystallin, the conformations of the initial structure for quantum chemical calculations were constructed. Catalyst molecules and ions used in this study abundantly exist in vivo and have been suggested to catalyze post-translational modification of proteins [41,42,43,44,45,46,47,48,49]. Energy minima and transitional state (TS) geometries were optimized without any constrains by the density functional theory (DFT) with B3LYP functional and 6-31+G(d,p) basis set in the polarized continuum model (PCM). This method was used in many studies for the reaction mechanisms of post-translational modifications [41,42,43,44,45,46,47,48,49]. Vibrational frequency calculations were performed for all the optimized geometries to obtain the zero-point energy and to confirm them with no imaginary frequency (for energy minima) or a single imaginary frequency (for TSs). IRC calculations from the TSs to confirm the energy minima connected by each TS. In addition, the single-point energies of all optimized geometries were calculated by MP2/6-311+G(2d,2p) level. All of the quantum chemical calculations were performed using Gaussian16 [59].

## 4. Conclusions

In this study, we computationally investigated the effects of main-chain conformations of Asn residues on the deamidation reaction. MD simulations and quantum chemical calculations demonstrated that Asn deamidation more easily occurs in the syn conformation than in the anti conformation. The activation energy in the phosphate-catalyzed reaction of the syn conformation (89.3 kJ·mol^−1^) was significantly lower than that of the anti conformation (111 kJ·mol^−1^). Therefore, main-chain conformation is considered to be one of the important factors determining deamidation rate. There is no need for flexibility of the main chain as long as the main-chain conformation is suitable for succinimide formation. Specifically, the dihedral angles *φ* and *φ*_H_ are important for determining the deamidation rate of Asn residues. These dihedral angles may be available for prediction of the deamidation rate. On the other hand, common features for *χ* of frequently or infrequently deamidated Asn residues were not discovered in this study. As side chains exposed to the surface of protein structures are more flexible than the main chains, *χ* is assumed to be difficult to use as a parameter for predicting Asn deamidation rate. Our results suggest the lifetime of peptide/protein drugs can be extended by excluding syn-conformation Asn residues. In a previous study, the activation barriers of Asp and isoAsp formation from succinimide were shown to be 22.4 kcal·mol^−1^ (93.6 kJ·mol^−1^) and 22.5 kcal mol^−1^ (94.1 kJ·mol^−1^), respectively [1]. These values are higher than that of succinimide formation from the syn conformation, and lower than from the anti conformation. Therefore, in the syn conformation, succinimide formation might not be the rate-determining step of Asp/isoAsp formation from Asn residues.

Our results also suggest that H_2_PO_4_^−^ and HCO_3_^−^ ions are the principle catalysts for Asn deamidation in vivo, and that an increase in concentration of these anions in cells can accelerate Asn deamidation in proteins. Therefore, computational results in this study suggest the possibility of mitigating age-related diseases by regulating the concentration of H_2_PO_4_^−^ and HCO_3_^−^ ions.

## Figures and Tables

**Figure 1 ijms-21-07035-f001:**
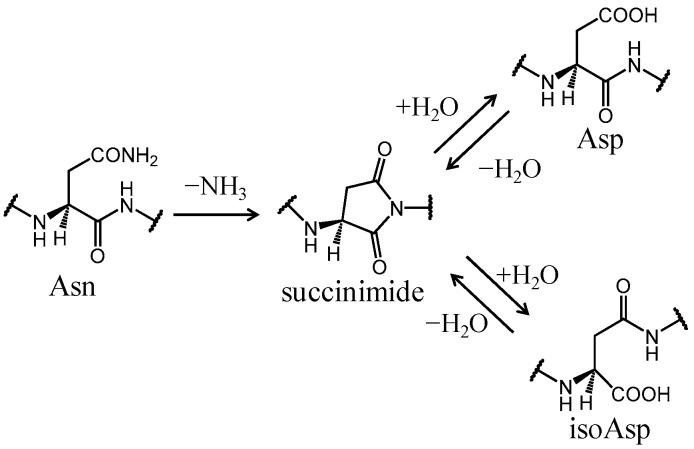
Succinimide-mediated deamidation pathways of asparagine (Asn) residues.

**Figure 2 ijms-21-07035-f002:**
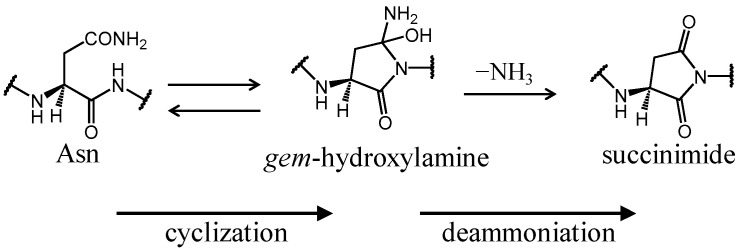
Proposed two-step succinimide formation pathway of Asn residues.

**Figure 3 ijms-21-07035-f003:**
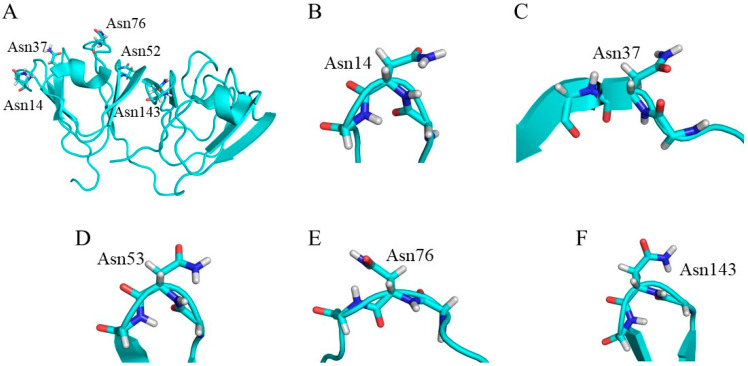
Conformation of Asn residues in γS-crystallin. (**A**) Experimentally determined structure of γS-crystallin. (**B**–**F**) Conformation of Asn residues in γS-crystallin.

**Figure 4 ijms-21-07035-f004:**
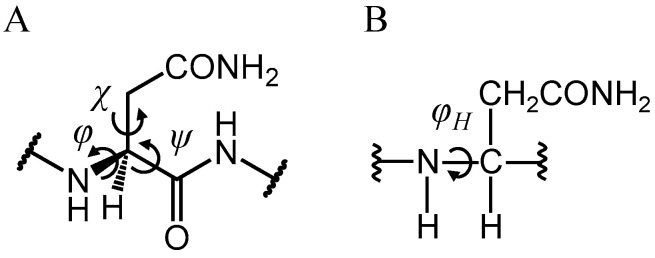
Dihedral angles used in this study to analyze the main-chain conformationof Asn residues (A for *φ*, *ψ*, and *χ,* B for *φ*_H_).

**Figure 5 ijms-21-07035-f005:**
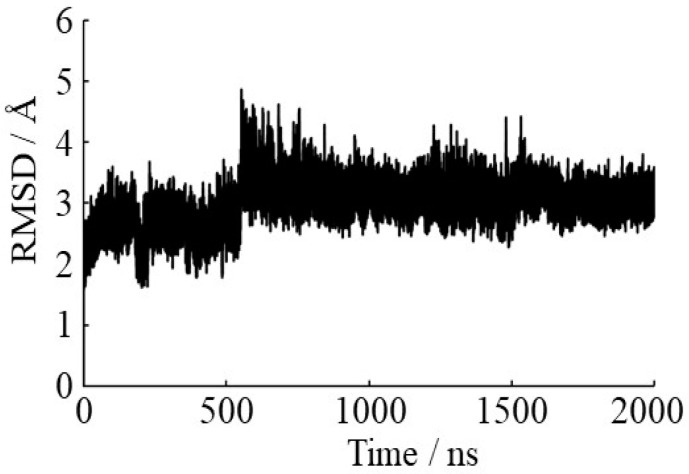
Root mean square deviation (RMSD) plots for the main-chain atoms of γS-crystallin.

**Figure 6 ijms-21-07035-f006:**
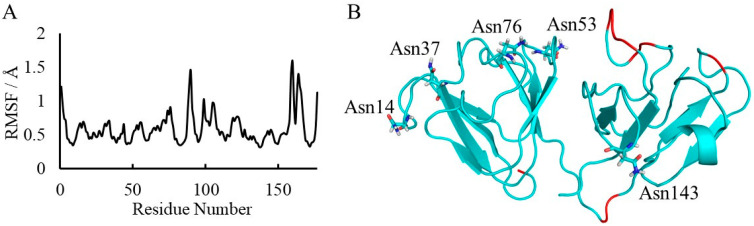
Flexibility of the backbone of γS-crystallin. (**A**) The root mean square fluctuation (RMSF) plot of γS-crystallin. (**B**) The final structure of γS-crystallin obtained by MD simulations. The segments of backbone where RMSF values are >1.0 Å are shown in red.

**Figure 7 ijms-21-07035-f007:**
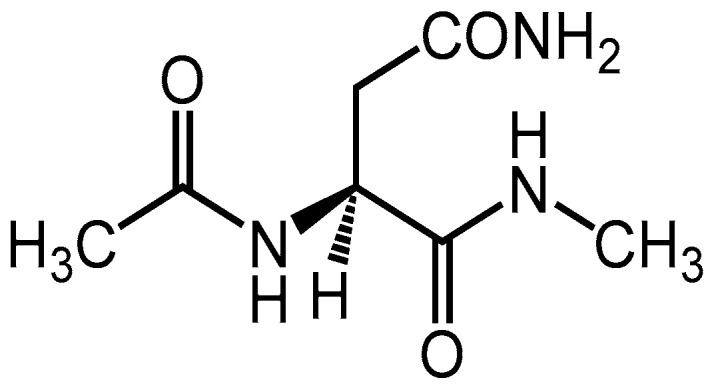
The structure of the model compound.

**Figure 8 ijms-21-07035-f008:**
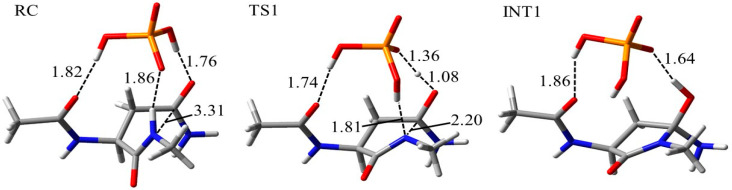
Optimized geometries of the cyclization step for the syn conformation in the phosphate-catalyzed reaction. The carbon, nitrogen, oxygen, and phosphorus atoms are illustrated in gray, blue, red, and orange, respectively. Selected interatomic distances are in units of Å.

**Figure 9 ijms-21-07035-f009:**
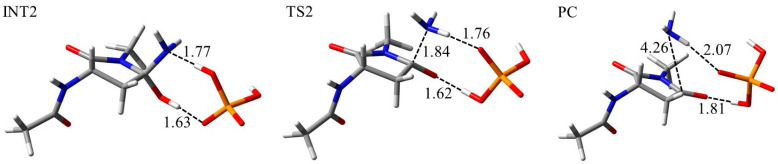
Optimized geometries of the deammoniation step for the syn conformation in phosphate-catalyzed reaction. The carbon, nitrogen, oxygen, and phosphorus atoms are illustrated in gray, blue, red, and orange, respectively. Selected interatomic distances are in units of Å.

**Figure 10 ijms-21-07035-f010:**
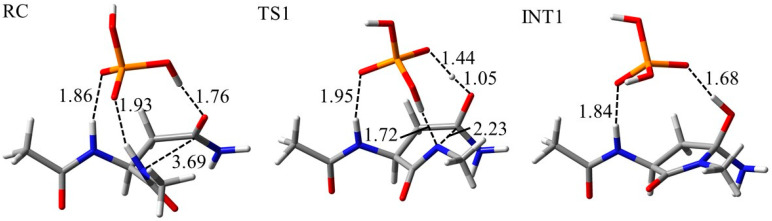
Optimized geometries of the cyclization step for the anti conformation in the phosphate-catalyzed reaction. The carbon, nitrogen, oxygen, and phosphorus atoms are illustrated in gray, blue, red, and orange, respectively. Selected interatomic distances are in units of Å.

**Figure 11 ijms-21-07035-f011:**
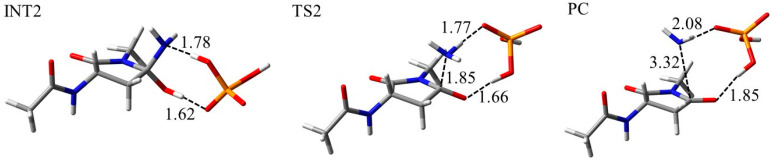
Optimized geometries of the deammoniation step for the anti conformation in the phosphate-catalyzed reaction. The carbon, nitrogen, oxygen, and phosphorus atoms are illustrated in gray, blue, red, and orange, respectively. Selected interatomic distances are in units of Å.

**Figure 12 ijms-21-07035-f012:**
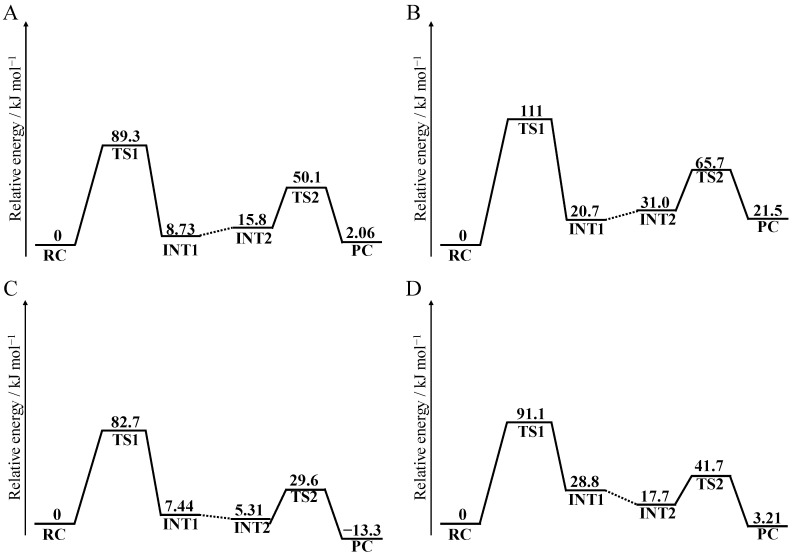
Relative energy profiles of the phosphate- and carbonate-catalyzed reaction calculated using the MP2/6-311+G(2d,2p)//B3LYP/6-31+G(d,p) level. Entire energy profiles of (**A**) syn conformation in the phosphate-catalyzed reaction, (**B**) anti conformation in the phosphate-catalyzed reaction, (**C**) syn conformation in the carbonate-catalyzed reaction, and (**D**) anti conformation in the carbonate-catalyzed reaction are shown.

**Table 1 ijms-21-07035-t001:** Dihedral angles (degrees) of Asn residues in the experimentally determined structure.

	Dihedral Angle/Degree			
Residue	*φ*	*ψ*	*χ*	*φ* _H_
Asn14	49.8	26.1	−41.9	−13.2
Asn37	−125	155	−68.8	173
Asn53	79.8	−4.29	−59.9	19.4
Asn76	107	160	−166	43.6
Asn143	70.9	19.5	−54.4	10.4

**Table 2 ijms-21-07035-t002:** Dihedral angles (degrees) of Asn residues in the final structure of the molecular dynamics (MD) simulation.

	Dihedral Angle/Degree			
Residue	*φ*	*ψ*	*χ*	*φ* _H_
Asn14	65.49	31.47	−52.05	−7.31
Asn37	−76.63	−17.67	−79.28	−154.86
Asn53	46.48	44.84	−171.13	−38.94
Asn76	−152.31	154.04	55.14	155.48
Asn143	59.88	43.14	−43.95	−13.20

**Table 3 ijms-21-07035-t003:** Percent of syn/anti conformation for the last 10 ns of the MD simulation.

	Asn14	Asn37	Asn53	Asn76	Asn143
syn	95.8	0	96.9	0	99.1
anti	0	50.4	0	70.8	0

**Table 4 ijms-21-07035-t004:** Dihedral angles (degrees) of the optimized geometries for the syn conformation in the phosphate-catalyzed reaction.

	Dihedral Angle/Degree			
	*φ*	*ψ*	χ	*φ* _H_
RC	71.04	−105.2	−174.0	4.917
TS1	57.74	−136.4	168.6	−7.248
INT1	55.41	−142.0	149.5	−9.097
INT2	54.72	−141.2	146.0	−7.442
TS2	54.95	−141.6	146.2	−6.697
PC	54.74	−140.5	139.3	−6.901

**Table 5 ijms-21-07035-t005:** Dihedral angles (degrees) of the optimized geometries for the anti conformation in the phosphate-catalyzed reaction.

	Dihedral Angle/Degree			
	*φ*	*ψ*	*χ*	*φ* _H_
RC	−83.20	−22.74	138.2	−151.8
TS1	−111.3	−109.5	144.8	−171.0
INT1	−117.0	−133.4	137.1	178.6
INT2	−114.2	−140.1	145.9	−179.0
TS2	−113.7	−142.7	147.2	−179.1
PC	−113.6	−139.6	140.0	−178.6

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
