# Peer review of "Mechanisms of Deamidation of Asparagine Residues and Effects of Main-Chain Conformation on Activation Energy"

_ijms, 2020, doi:10.3390/ijms21197035_

Round 1

Reviewer 1 Report

This is an interesting study of the deamination of asparagine in relation to human alpha-crystallin aging. The Authors used canonical MD simulations of crystallin to determine the flexibility and conformations of the asparagine residues and then performed detailed DFT study, with calculated more accurate single-point energies at the MP2 level, of the reaction mechamisms catlyzed by the dihydrgogen phosphate and hydrogen carbonate anions. The solvent was accounted for in both the MD and QM calculations. They found that the syn conformation is more liable to deamination. The calculated reaction barriers are reasonable and consistent with long timescale of the reactions under study. The paper is cleary written. I have the following remarks and suggestions:

(1) The mechanism involving catalysis by hydrogen carbonate (bicarbonate) is said to be presented in the SI. However, I could not find SI anywhere and, therefore, am unable ot assess this part of the study. Only the reaction barriers are quoted in line 277 (page 9). On the other hand, it seems not to take much more space to include at least the energy levels of the reactions mediated by bicarbonate in addtional panels of Figure 12.

(2) The study seems to extend only until the cyclic succinimide, not to the final products which are aspartate and iso-aspartate (the last step of the complete reaction sketched in Figure 1). While this step does not distinguish the syn- and anti-conformations of the asparagine, it seems worthwhile to estimate the barriers of the last step or quote the respective values, if they are available from other studies.

(3) Some comparison, at least qualitative, with the experimental data (Gibbs free energy barriers, rate constants) should be presented.

Author Response

Thank you very much for your valuable comments. I agree that several points can be clarified and we have modified the article accordingly. Specific responses are provided below. Regarding the correction point of the text is shown in red in the file with marked manuscript revision.

Reviewer1

This is an interesting study of the deamination of asparagine in relation to human alpha-crystallin aging. The Authors used canonical MD simulations of crystallin to determine the flexibility and conformations of the asparagine residues and then performed detailed DFT study, with calculated more accurate single-point energies at the MP2 level, of the reaction mechamisms catlyzed by the dihydrgogen phosphate and hydrogen carbonate anions. The solvent was accounted for in both the MD and QM calculations. They found that the syn conformation is more liable to deamination. The calculated reaction barriers are reasonable and consistent with long timescale of the reactions under study. The paper is cleary written. I have the following remarks and suggestions:

(1) The mechanism involving catalysis by hydrogen carbonate (bicarbonate) is said to be presented in the SI. However, I could not find SI anywhere and, therefore, am unable to assess this part of the study. Only the reaction barriers are quoted in line 277 (page 9). On the other hand, it seems not to take much more space to include at least the energy levels of the reactions mediated by bicarbonate in additional panels of Figure 12.

Response:

We sincerely apologize for the mistake in submission of the SI. We have now uploaded the SI with the revised manuscript; please review. In addition, we appreciate the reviewer’s advice concerning Figure 12. Per the reviewer's suggestion, relative energy profiles of the carbonate-catalyzed reaction were added to Figure 12, and reference to the figure in the main text has been modified.

(2) The study seems to extend only until the cyclic succinimide, not to the final products which are aspartate and iso-aspartate (the last step of the complete reaction sketched in Figure 1). While this step does not distinguish the syn- and anti-conformations of the asparagine, it seems worthwhile to estimate the barriers of the last step or quote the respective values, if they are available from other studies.

Response:

      Experimental activation barriers for Asp and isoAsp formation from succinimide in phosphate buffer have been reported to be 22.4 kcal mol−1 (93.6 kJ mol−1) and 22.5 kcal mol−1 (94.1 kJ mol−1), respectively [Ref.1]. These values are higher than that of succinimide formation from the syn conformation, while they are lower than that from anti-conformation. Accordingly the sentences below have been added to the Conclusion.

“In a previous study, the activation barriers of Asp and isoAsp formation from succinimide were shown to be 22.4 kcal mol−1 (93.6 kJ mol−1) and 22.5 kcal mol−1 (94.1 kJ mol−1), respectively [1]. These values are higher than that of succinimide formation from the syn conformation, and lower than from the anti-conformation. Therefore, in the syn-conformation succinimide formation might not be the rate determining step of Asp/isoAsp formation from Asn residues.”

(3) Some comparison, at least qualitative, with the experimental data (Gibbs free energy barriers, rate constants) should be presented.

Response:

         We agree that a comparison with experimental data is needed. We compared the calculated activation barriers with experimental values in model peptides. Calculated activation barriers of the syn conformation were 89.3 and 82.7 kJ mol−1 in phosphate- and carbonate-catalyzed reactions, respectively, and the experimentally determined values were 80–100 kJ mol−1 [Ref.6,7]. These values are in agreement. Notably the value in phosphate buffer (pH 7.5) is 21.7 kcal mol l−1 (90.7 kJ mol−1) [Ref.7], almost identical to our calculated activation barrier of the phosphate-catalyzed reaction. On the other hand, a frequency factor is required for determination of the rate constant, but this value cannot be obtained by the quantum chemical calculations employed in this study. Comparison of activation barriers is addressed in Section 2.4 (lines 326–328 and 332–334). Please confirm that they address your concerns.

Reviewer 2 Report

Spontaneous deamidation of Asn in proteins may cause different illnesses, therefore, the knowledge of the mechanism of the Asn rearrangement to Asp and isoAsp under physiological condition is important. There are many experimental and theoretical studies to understand the mechanism of deamidation. The authors followed the later one and they could properly prove the importance of phosphate and carbonate ions in catalysis the rearrangement through succinimide formation. The manuscript is well written and the results are well presented. According to my opinion the MD calculation might provide solid background for determination of the mechanism, but the experimental studies are usually needed to confirm the results. Therefore, I miss from the introduction part two recently published NMR studies of deamidation procedure that closely related to this publication. I think they had to be referred.

  1. Enyedi KN, et al. Development of cyclic NGR peptides with thioether linkage: structure and dynamics determining deamidation and bioactivity. J Med Chem 2015, 58, 4, 1806–1817
  2. Láng A, et al. Off-pathway 3D-structure provides protection against spontaneous Asn/Asp isomerization: shielding proteins Achilles heel. Quarterly Reviews of Biophysics Volume 532020 , e2

Author Response

Spontaneous deamidation of Asn in proteins may cause different illnesses, therefore, the knowledge of the mechanism of the Asn rearrangement to Asp and isoAsp under physiological condition is important. There are many experimental and theoretical studies to understand the mechanism of deamidation. The authors followed the later one and they could properly prove the importance of phosphate and carbonate ions in catalysis the rearrangement through succinimide formation. The manuscript is well written and the results are well presented. According to my opinion the MD calculation might provide solid background for determination of the mechanism, but the experimental studies are usually needed to confirm the results. Therefore, I miss from the introduction part two recently published NMR studies of deamidation procedure that closely related to this publication. I think they had to be referred.

  • Enyedi KN, et al. Development of cyclic NGR peptides with thioether linkage: structure and dynamics determining deamidation and bioactivity. J Med Chem 2015, 58, 4, 1806–1817

  • Láng A, et al. Off-pathway 3D-structure provides protection against spontaneous Asn/Asp isomerization: shielding proteins Achilles heel. Quarterly Reviews of Biophysics Volume 532020 , e2

Response:

            We appreciate the reviewer’s advice and for alerting us to the two studies. The sentences below refer to those studies and were added to Introduction, along with revision of the following sentence (lines 86–95). We would like you to confirm that they are responsive to your concerns.

“In cyclic peptides containing the Asn–Gly sequence, the deamidation rate is higher in the peptide with the shorter NN+1–Cγ distance [39,40]. In addition, deamidation is facilitated when the NN+1–Cγ–Oγ angle is <128°. However, deamidation rate is low in a cyclic peptide satisfying these parameters (Lys–Asn–Gly–Arg–Glu–NH2). Therefore, although these parameters are key in determining deamidation rate, other factors are also important. Those parameters might be difficult to apply to the flexible side chain of Asn residues in protein structures. Therefore, the relevance for deamidation rate of the NN+1–Cγ distance and dihedral angles in crystal structures is unclear for calculations of protein dynamics. ”

Round 2

Reviewer 1 Report

The Authors have addressed my critisicsm satisfactorily and I now recommend the manuscript for publication.